

# Methane paradox in tropical lakes? Sedimentary fluxes rather than water column production in oxic waters sustain methanotrophy and emissions to the atmosphere

Cédric Morana[1,2]*, Steven Bouillon[1], Vimac Nolla-Ardèvol[1], Fleur A.E. Roland[2], William Okello[3], Jean-Pierre Descy[2], Angela Nankabirwa[3], Erina Nabafu[3], Dirk Springael[1], Alberto V. Borges[2].

[1]Department of Earth & Environmental Sciences, KU Leuven, Belgium
[2]Chemical Oceanography Unit, Université de Liège, Belgium
[3]Limnology Unit, National Fisheries Resources Research Institute, Uganda

*Correspondence to*: Cédric Morana (Cedric.Morana@kuleuven.be)

**Abstract.** Despite growing evidence that methane ($CH_4$) formation could also occur in well-oxygenated surface freshwaters, its significance at the ecosystem scale is uncertain. Empirical models based on data gathered at high latitude predict that the contribution of oxic $CH_4$ increases with lake size and should represent the majority of $CH_4$ emissions in large lakes. However, such predictive models could not directly apply to tropical lakes which differ from their temperate counterparts in some

fundamental characteristics, such as year-round elevated water temperature. We conducted stable isotope tracer experiments which revealed that oxic $CH_4$ production is closely related to phytoplankton metabolism, and is a common feature in five contrasting African lakes. Nevertheless, methanotrophic activity in surface waters and $CH_4$ emissions to the atmosphere were predominantly fuelled by $CH_4$ generated in sediments and physically transported to the surface. Indeed, measured $CH_4$ bubble dissolution flux and diffusive benthic $CH_4$ flux were several orders of magnitude higher than $CH_4$ production in surface waters.

Microbial $CH_4$ consumption dramatically decreased with increasing sunlight intensity, suggesting that the freshwater "$CH_4$ paradox" might be also partly explained by photo-inhibition of $CH_4$ oxidizers in the illuminated zone. Sunlight appeared as an overlooked but important factor determining the $CH_4$ dynamics in surface waters, directly affecting its production by photoautotrophs and consumption by methanotrophs.

## 1. Introduction

Emissions from inland waters are an important component of the global $CH_4$ budget (Bastviken et al. 2011), in particular from tropical latitudes (Borges et al. 2015). While progress has been made in evaluating the $CH_4$ emission rates, much less attention has been given to the underlying microbial production (methanogenesis) and loss (methane oxidation) processes. It is generally assumed that $CH_4$ in lakes originates from the degradation of organic matter in anoxic sediments. Because most methanogens are considered to be strict anaerobes and net vertical diffusion of $CH_4$ from anoxic bottom waters is often negligible (Bastviken

et al. 2003), physical processes of $CH_4$ transport from shallow sediments are usually invoked to explain patterns of local $CH_4$ concentration maximum in surface waters (Encinas-Fernandez et al. 2016, Peeters et al. 2019, Martinez-Cruz et al. 2020). Indeed, $CH_4$-rich pore water is regularly released from littoral sediment into the water column during resuspension events associated with surface waves (Hofmann et al. 2010).

The view that $CH_4$ is formed under strictly anaerobic conditions has been challenged by several recent studies which proposed

that acetoclastic methanogens directly attached to phytoplankton cells are involved in epilimnetic $CH_4$ production (Grossart et al. 2011, Bogard et al. 2014), and are responsible of distinct near-surface peaks of $CH_4$ concentration in certain thermally stratified, well-oxygenated waterbodies (Tang et al. 2016). It has also been showed that Cyanobacteria (Bizic et al. 2020) and widespread marine phytoplankton (Klintzsch et al. 2019) are able to release substantial amount of $CH_4$ during a culture study, and this $CH_4$ production mechanism might be linked to photosynthesis. From a model-based approach, epilimnetic $CH_4$

production was shown to sustain most of the $CH_4$ oxidation in 14 Canadian lakes (DelSontro et al. 2018), and would even





represent up to 90% of the CH$_4$ emitted from a temperate lake (Donis et al. 2017). Further, empirical models based on data gathered in boreal and temperate lakes predict that the contribution of oxic CH$_4$ increases with lake size (Gunthel et al. 2019) and should represents the majority of CH$_4$ emissions in lakes larger than 1 km². Still, aerobic CH$_4$ production has so far only been documented in temperate and boreal lakes so that such predictive models could not directly apply to tropical lakes which

differ from their temperate counterparts in some fundamental characteristics, such as year-round elevated water temperature. Among others, primary production, methanogenic and methanotrophic activities, and cyanobacterial dominance are potentially much higher in tropical lakes due to favorable temperature (Lewis 1987, Kosten et al. 2012). It has also been shown that CH4 emissions are positively related to temperature at the ecosystem scale (Yvon-Durocher et al. 2014)

Here, we tested the hypothesis that phytoplankton metabolism could fuel CH$_4$ production in well-oxygenated waters in five

contrasting tropical lakes in East Africa covering a wide range of size, depth, and productivity (L. Edward, L. George, L. Katinda, L. Nyamusingere and L. Kyambura). Phytoplankton activity could provide diverse substrates required for CH$_4$ production mediated by methanogenic Archaea, or alternatively CH$_4$ could be directly released by phytoplankton cells. Additionally, the significance of epilimnetic CH$_4$ production at the scale of the aquatic ecosystem was assessed by quantifying CH4 release from sediments, CH$_4$ production and oxidation rates in the water column, and CH$_4$ diffusive and ebullitive

emissions to the atmosphere.

## 2. Material and methods

### 2.1. Site description

The sampled lakes cover a wide range of size (<1 to 2300 km²), maximum depth (3-117 m), mixing regimes, phytoplankton biomass and primary productivity (Table S1). Oligotrophic L. Kyamwinga (-0.18054°N, 30.14625°E) and eutrophic L. Katinda

(-0.21803°N, 30.10702°E) are stratified, small but deep tropical lakes located in Western Uganda. Neighboring L. Nyamusingere (-0.284364°N, 30.037635°E) is a small but shallow and polymictic eutrophic lake. L. George is a larger (250 km²), hypereutrophic, shallow lake located at the equator (-0.02273°N, 30.19724°E). A single outlet (Kazinga Channel) flows from L. George to the neighboring Lake Edward (-0.28971°N, 29.73327°E), a holomictic, mesotrophic large lake (2325 km²). Water samples from pelagic stations of L. Katinda, L. George (2km offshore) and L. Edward (15 km offshore) were collected

in April 2017 (rainy season) and January 2018 (dry season). Pelagic sites of L. Kyamwinga and L. Nyamusingere were sampled only once, in April 2017 and January 2018, respectively.

### 2.2. Environmental setting of the study sites

Conductivity, temperature and dissolved oxygen concentration measurements were performed with a Yellow Spring Instrument EXO II multiparametric probe. Samples for particulate organic carbon (POC) concentration were collected on glass

fiber filters (0.7 μm nominal pore size) and analyzed with an elemental analyzer coupled to an isotope ratio mass spectrometer (EA-IRMS) (Morana et al. 2015). Pigment concentrations were determined by high performance liquid chromatography (Descy et al. 2016) after filtration of water samples through glass fiber filters (0.7 μm nominal pore size).

Water samples for determination of dissolved CH$_4$ concentration were transferred with tubing from the Niskin bottle to 60 ml borosilicate serum bottles that were poisoned with 200μL of a saturated solution of HgCl$_2$, closed with a butyl stopper

and sealed with an aluminum cap. The concentrations of dissolved CH$_4$ was measured with the headspace equilibration technique (20 ml headspace) using a gas chromatograph with flame ionization detection (GC-FID, SRI8610C).

Samples for $\delta^{13}$C-CH$_4$ determination were collected in 60 ml serum bottles following the same procedure than samples for CH$_4$ concentration determination. In the laboratory, $\delta^{13}$C-CH$_4$ was measured as described in Morana et al. (2015). Briefly, a 20ml helium headspace was created in the serum bottles, then samples were vigorously shaken and left to equilibrate

overnight. The sample gas was flushed out through a double-hole needle and purified of non-CH$_4$ volatile organic compounds



in a liquid $N_2$ trap, $CO_2$ and $H_2O$ were removed with a soda lime and a magnesium perchlorate traps, and the $CH_4$ was converted to $CO_2$ in an online combustion column similar to that in an elemental analyzer (EA). The resulting $CO_2$ was subsequently preconcentrated in a custom-built cryo-focussing device by immersion of a stainless-steel loop in liquid $N_2$, passed through a micro-packed GC column (HayeSep Q 2 m, 0.75mm ID; Restek), and finally measured on a Thermo Scientific Delta V

Advantage isotope ratio mass spectrometer (IRMS). $CO_2$ produced from certified reference standards for $\delta^{13}C$ analysis (IAEA-CO1 and LSVEC) were used to calibrate $\delta^{13}C$-$CH_4$ data. Reproducibility of measurement estimated based on duplicate injection of a selection of samples was typically better than 0.5 ‰, or better than 0.2‰ when estimated based on multiple injection of standard gas.

### 2.3. Diffusive $CH_4$ flux calculation

Surface $CH_4$ concentrations were used to compute the diffusive air-water $CH_4$ fluxes ($FCH_4$) according to eq. (1):

$$FCH_4 = k \times \Delta CH_4 \tag{1}$$

Where k is the gas transfer velocity of $CH_4$ computed from wind speed (Cole & Caraco 1998) and the Schmidt number

of $CH_4$ in freshwater (Wanninkhof 1992), and $\Delta CH_4$ is the air-water gradient. Wind speed data were acquired with a Davis Instruments meteorological station located in Mweya peninsula (0.11°S 29.53°E).

### 2.4. $CH_4$ ebullition flux

     $CH_4$ ebullition flux was investigated in In L. Edward, George, and Nyamusingere only. Bubble traps made with an inverted funnel (24 cm diameter) connected to a 60 ml syringe were deployed for a period between 24 h and 48 h at 0.5 m

below the water surface (4 replicates). Measurements were performed at sites with water depth of 20 m, 2.5 m and 3 m for L. Edward, George and Nyamusingere, respectively. After measuring the gas volume collected within the trap during the sampling period, the gas bubbles were transferred in a tightly closed 12 ml Exetainer vial (Labco) for subsequent analysis of their $CH_4$ concentration. Variability of the gas volume in the 4 replicates was less than 10%. We used the SiBu-GUI software (McGinnis et al. 2006, Greinert et al. 2009) to correct for gas exchange within the water column during the rise of bubbles and

thus obtained the $CH_4$ ebullition and $CH_4$ bubble dissolution fluxes. Calculations were made following several scenarios: two extreme bubble-size scenarios considering a release of many small (3 mm diameter) bubbles or fewer large (10 mm) bubbles, and an intermediate scenario of release of 6 mm diameter bubbles.

### 2.5. $CH_4$ flux across the sediment-water interface

     $CH_4$ flux across the sediment-water interface was determined from short-term intact core incubations in L. Edward,

L. George and L. Nyamusingere only. $CH_4$ flux was quantified from the change of $CH_4$ concentration in overlying waters at 5 different time steps, every 2 hours. Briefly, in every lake, 2 sediment cores (6 cm wide; ~ 30cm sediment and 30cm of water) were collected taking care to avoid disturbance at the sediment-water interface. Cores were kept in the dark until back in the laboratory, typically 6h later. Overlying water was carefully removed and replaced by bottom lake water filtered through 0.2μm polycarbonate filters (GSWP, Millipore) in order to remove water column methanotrophs. It was then degassed with

helium during 20 minutes in order to remove background $O_2$ and $CH_4$, and gently returned in the core tubes, on top of the sediments. Core tubes were tightly closed with a thick rubber stopper equipped with two sampling valves. A magnetic stirrer placed ~ 10 cm above the sediments was allowed to rotate gently in order to homogenize the overlying water layer during the incubation. At each time step, 60 ml of overlying water was sampled by connecting a syringe to the first sampling valve while an equivalent volume of degassed water was allowed to flow through the second valve in order to avoid any pressure

disequilibrium. Subsamples of overlying water were transferred into a two 20 ml serum bottles filled without headspace and





poisoned with HgCl₂. Determination of the dissolved CH₄ concentration was performed with a GC-FID following the same procedure as described above.

**2.6. Primary production and N₂ fixation**

Primary production and N₂ fixation rates were determined from dual stable isotope photosynthesis-irradiance experiments using NaH$^{13}$CO₃ (Eurisotop) and dissolved $^{15}$N₂ (Eurisotop) as tracers for incorporation of dissolved inorganic carbon (DIC) and N₂ into the biomass. The $^{15}$N₂ tracer was added dissolved in water (Mohr et al. 2010). Incident light intensity was measured by a LI-190SB quantum sensor during day time during the entire duration of the sampling campaign. At each station a sample of surface waters (500 ml) was spiked with the tracers (final $^{15}$N atom excess ~5%). Three subsamples were preserved with HgCl₂ in 12-mL Exetainers vials (Labco) for the determination of the exact initial $^{13}$C-DIC and $^{15}$N-N₂ enrichment. The rest of the sample was divided into nine 50-ml polycarbonate flasks, filled without headspace. Eight flasks were placed into a floating incubation device providing a range of light intensity (from 0 to 80% of natural light) using neutral density filter screen (Lee Filters). The last one was immediately amended with neutral formaldehyde (0.5% final concentration) and served as killed control sample. Samples were incubated *in situ* during 2 hours around mid-day just below the surface at lake surface temperature. After incubation, biological activity was stopped by adding neutral formaldehyde into the flasks, and the nine samples were filtered on pre-combusted GF/F filters when back in the lab. Glass fiber filters were decarbonated with HCl fumes overnight, dried, and their $\delta^{13}$C-POC and $\delta^{15}$N-PN values were determined with an EA-IRMS (Thermo FlashHT – delta V Advantage). For the measurement of the initial $^{15}$N₂ enrichment, a 2-ml helium headspace was created, and after 12h equilibration, a fraction of the headspace was injected into the above-mentioned EA-IRMS equipped with a Cu column warmed at 640°C and a CO₂ trap. Initial enrichment of $^{13}$C-DIC was also measured.

Photosynthetic (P$_i$) (Hama et al. 1983) and N₂ fixation (N₂fix$_i$) (Montoya et al. 1996) rates in individual bottles were calculated, and corrected for any abiotic tracer incorporation by subtraction of the killed control value. For each experiment, the maximum photosynthetic and N₂ fixation rates (P$_{max}$, N₂fix$_{max}$) and the irradiance at the onset of light saturation (I$_{k\_PP}$, I$_{k\_N2fix}$) were determined by fitting P$_i$ and N2fix$_i$ to the light intensity gradient provided by the incubator (I$_i$) using the equation (eq. 2) for photosynthesis activity (Vollenweider 1965) and (eq. 3) for N₂ fixation (Mugidde et al. 2003).

$$P_i = 2P_{max}\left[\frac{I_i/2I_{k\_PP}}{1+(I_i/2I_{k\_PP})^2}\right] \qquad (2)$$

$$N_2fix_i = 2N_2fix_{max}\left[\frac{I_i/2I_{k\_N2fix}}{1+(I_i/2I_{k\_N2fix})^2}\right] \qquad (3)$$

**2.7. Determination of CH₄ oxidation rates.**

CH₄ oxidation rates in surface waters (1m depth) were determined from the decrease of CH₄ concentrations measured during short (typically < 24h) time course experiments. Samples for CH₄ oxidation rate measurement were collected in 60 mL glass serum bottles filled directly from the Niskin bottle with tubing, left to overflow, and immediately closed with butyl stoppers previously boiled in milli-Q water, and sealed with aluminum caps. The first bottle was then poisoned with a saturated solution of HgCl₂ (100 µl) injected through the butyl stopper with a polypropylene syringe and a steel needle and corresponded to the initial CH₄ concentration at the beginning of the incubation (T0).

The remaining bottles were incubated in the dark, at in situ (~26°C) temperature during ~12h or ~24h except in L. George and Nyamusingere where the incubation was shorter (~6h). At 4 different times step one bottle was poisoned with 100 µL of HgCl₂ and stored in the dark until measurement of the CH₄ concentrations with the above-mentioned GC-FID. CH₄ oxidation rates were calculated as a linear regression of CH₄ concentrations over time (r² generally better than 0.80) during the course of the incubation.



**2.8. Sunlight inhibitory effect on CH₄ oxidation**

The influence of light intensity on methanotrophy was investigated in Lake Edward and Lake George by means of a stable isotope ($^{13}CH_4$) labelling experiment. For each experiment, 12 serum bottles (60 mL) were filled with lake surface waters (1m) as described above. All bottles were spiked with 100 µL of a solution of dissolved $^{13}CH_4$ (50 µmol L$^{-1}$ final concentration, 99% enrichment) added in excess. Half of the bottles were amended with 3-(3,4-dichlorophenyl)-1,1-dimethylurea (DCMU, 0.5 mg L$^{-1}$) in order to inhibit photosynthesis (Bishop 1958) and investigate the hypothetical inhibitory effect of dissolved $O_2$ production by phytoplankton. Two bottles were poisoned immediately with pH-neutral formaldehyde (0.5% final concentration) and served as killed controls. The ten others were incubated during 24h at 26°C in a floating device providing 5 different light intensities (from 0 to 80% of natural light using neutral density filter screens (Lee Filters). For every bottle at the end of the incubation, one 12-mL vial (Labco Exetainer) was filled with the water sample and preserved with 50 µL HgCl$_2$. The rest of the sample (~50 mL) was filtered on a precombusted GF/F filter for subsequent $\delta^{13}$C-POC measurement.

$\delta^{13}$C-DIC and $\delta^{13}$C-POC were determined with an EA-IRMS as described above. The methanotrophic bacterial production, defined at the CH$_4$-derived $^{13}$C incorporation rates into the POC pool was calculated as in eq. (4) (Morana et al. 2015):

$$MBP = \frac{POC_t \times (\%^{13}CPOC_t/\%^{13}CPOC_i)}{t \times (\%^{13}CCH_4/\%^{13}CPOC_i)} \tag{4}$$

Where POC$_t$ is the concentration of POC after incubation, $\%^{13}$C-POC$_t$ and $\%^{13}$C-POC$_i$ are the final and initial percentage of $^{13}$C in the POC, t is the incubation time and $\%^{13}$C-CH$_4$ is the percentage of $^{13}$C in CH$_4$ after the inoculation of the bottles with the tracer. Similarly, the methanotrophic bacterial respiration rates, defined as the CH$_4$-derived $^{13}$C incorporation rates into the DIC pool, were calculated as in eq. (5):

$$MBR = \frac{DIC_t \times (\%^{13}CDIC_t/\%^{13}CDIC_i)}{t \times (\%^{13}CCH_4/\%^{13}CDIC_i)} \tag{5}$$

Where DIC$_t$ is the concentration of DIC after the incubation, $\%^{13}$C-DIC$_t$ and $\%^{13}$C-DIC$_i$ are the final and initial percentage of $^{13}$C in DIC and $\%^{13}$C-CH$_4$ is the percentage of $^{13}$C in CH$_4$ after the inoculation of the bottles with the tracer.

Potential CH$_4$ oxidation rates (MOX) were calculated as the sum of MBP and MBR rates. The fraction (%) of MOX inhibited by light was calculated at every light intensity as (eq.6):

$$MOX_{inihibition}(\%) = (1 - MOX_i/MOX_{dark}) \times 100 \tag{6}$$

Where MOX$_i$ is the potential CH$_4$ oxidation for a given light treatment and MOX$_{dark}$ is the potential CH$_4$ oxidation in the dark.

**2.9. Determination of pelagic CH₄ production rates.**

Time course $^{13}$C tracer experiments were carried out in well oxygenated surface waters at every sampling site. Measurement of the isotopic enrichment of the CH$_4$ during this experiment allowed to estimate production rates of CH$_4$ issued from 3 different precursors: $^{13}$C-DIC (NaH$^{13}$CO$_3$), $^{13}$C$_{(1,2)}$-acetate and $^{13}$C$_{methyl}$-methionine. Serum bottles (60 ml) were spiked with 1 ml of $^{13}$C tracer solution, or with an equivalent volume of distilled water for the control treatment. NaH$^{13}$CO$_3$ was added in the bottles at a tracer level (less than 5% of ambient HCO$_3^-$ concentration) while $^{13}$C$_{(1,2)}$-acetate and $^{13}$C$_{methyl}$-methionine were added largely in excess (>99% of ambient concentration). Therefore, we assume the CH$_4$ production rates measured from $^{13}$C-DIC could be representative of in-situ rates, but the production rates measured from $^{13}$C-acetate and $^{13}$C-methionine should

instead be viewed as potential rates. The exact amount of $^{13}$C-DIC added in the bottles was determined filling a borosilicate 12 ml exetainer vials preserved and analysed for $\delta^{13}$C-DIC as described above.

The control bottles and the bottles amended with the different $^{13}$C tracer were incubated under constant temperature conditions (26°C) following three different treatments : (1) one third were incubated under constant light (PAR of ~ 200 µmol photon m$^{-2}$ s$^{-1}$), (2) another third were incubated under the same light intensities conditions but were first amended with DCMU (0.5 mg L$^{-1}$ ; final concentration), an inhibitor of photosynthesis, (3) and the last third were incubated in the dark.

At each time step (typically every 6-12h, 5-time steps), the biological activity was stopped by adding 100 µL of a saturation solution of HgCl$_2$. Bottles were kept in the dark until CH$_4$ concentration measurement and $\delta^{13}$C-CH$_4$ determination as described above.

The term CH$_{4\_prod}$ (nmol L$^{-1}$ h$^{-1}$) defined as the amount of CH$_4$ produced from a specific tracer during a time interval t (h), was calculated following this equation (eq. 7) derived from Hama et al. (1983):

$$CH_{4\_prod} = \frac{CH_{4\_t} \times (\%^{13}CCH_{4\_t}/\%^{13}CCH_{4\_i})}{t \times (\%^{13}Ctracer/\%^{13}CCH_{4\_i})} \tag{7}$$

Where CH$_{4\_t}$ and %$^{13}$CCH$_{4\_t}$, represent the CH$_4$ concentration (nmol L$^{-1}$) and the %$^{13}$C atom of the CH$_4$ pool at a given time step, respectively. %$^{13}$CCH$_{4\_i}$ represent the %$^{13}$C atom of the pool of CH$_4$ at the beginning of the experiment. %$^{13}$Ctracer represent the %$^{13}$C atom of the isotopically enriched pool of the precursor molecule tested (NaHCO$_3$, methionine or acetate, depending of the treatment). %$^{13}$C-tracer was assumed constant during the full course of the incubation given the high concentration of ambient DIC in the sampled lakes (~ 2 mmol L$^{-1}$ in L. George, > 6 mmol L$^{-1}$ in the other lakes) and that acetate and methionine were spiked in large excess (>99%).

**2.10. DNA extraction**

Surface water sample for DNA analysis (between 1 L and 0.15 L, depending on the biomass) were first filtered through 5.0 $\mu$m pore size polycarbonate filters (Millipore). The eluent was then subsequently filtered through 0.2 $\mu$m pore size polycarbonate filters (Millipore) to retain free living prokaryotes. Filters were stored frozen (-20°C) immerged in a lysis buffer until processing in the laboratory. Total DNA was extracted from the 0.2 $\mu$m and 5.0 $\mu$m 47 mm filters using DNeasy PowerWater kit (Qiagen) following the manufacturer's instructions. Quality and quantity of the extracted DNA were estimated using the NanoDrop ND-1000 spectrophotometer (ThermoFisher) and the Qubit 3.0 fluorometer (Life technology). Extracted DNA was stored at -20 °C until further use.

**2.11. Quantification of mcrA via qPCR**

Quantification of *mcrA* gene copies was performed by quantitative PCR (qPCR) on the total extracted DNA. The used primer pair consisted of forward primer *qmcrA-F* 5'-TTCGGTGGATCDCARAGRGC-3'and *qmcrA-R* 5'-GBARGTCGWAWCCGTAGAATCC-3' (Denman et al. 2007). The reaction mixture contained 3 µL of total community DNA extract, 7.5 µL ABsolute qPCR SYBR Green Mix (ThermoFisher, Cat. AB1158B), 0.3 µL of 10 µM forward primer *mcrF*, 0.3 µL of 10 µM reverse primer *mcrR*, 1.5 µL of a 1% w/v Bovine Serum Albumin solution (Amersham Bioscience) and 2.4 µL of Nuclease/DNA-free water. The qPCR was performed in a Rotorgene 3000 (Corbett Research) using the following conditions: 95 °C (15 min) followed by 40 cycles of 20 s at 95 °C, 20 s at 58 °C, 20 s at 72 °C and a final extension step of 5 s at 80 °C. Standard curves were prepared from serial dilutions of a prequantified *mcrA* PCR fragment amplified using primers mcrF and mcrR from a plasmid extract carrying the complete *mcrA* gene using concentrations ranging from 1x10$^2$ to 1x10$^8$ copies µL$^{-1}$. Samples were analyzed in triplicates.

**2.12. 16S rRNA gene amplicon sequencing**





Sequencing of the 16S rRNA gene was done on the total extracted DNA to assess community composition. 16S rRNA gene sequencing was done with the Illumina MiSeq v3 Chemistry following the "16S Metagenomic Sequencing Library Preparation" protocol with the following universal 16S rRNA gene primers targeting the V4 region, forward UniF/A519F-(S-D-Arch-0519-a-S-15) 5'-CAGCMGCCGCGGTAA-3' and reverse UniR/802R-(S-D-Bact-0785-b-A-18) 5'-TACNVGGGTATCTAATCC-3' (Klindworth et al. 2013). Sequenced read quality was checked using FastQC v0.11 (https://www.bioinformatics.babraham.ac.uk/projects/fastqc/). Short reads were trimmed to 250 bp with FastX Toolkit v0.0.13 (http://hannonlab.cshl.edu/fastx_toolkit/) in order to remove trailing Ns and low quality bases. Operational Taxonomical Units (OTU) for each analyzed sample were obtained from the quality trimmed reads using mothur v1.39.5 (Kozich et al. 2013) and following the online MiSeq SOP (https://mothur.org/wiki/MiSeq_SOP - accessed April 2018) using the Silva v128 16S rRNA database with the following parameters: *maxambig = 0 bp*; *maxlength = 300 bp*; *maxhomop = 8*; and classify OTUs to 97% identity. Generated OTU table was used to calculate relative abundances of each OTU per sample.

### 3. Results and discussion

#### 3.1. Patterns of phytoplankton biomass and dissolved CH$_4$

The sampled lakes cover a wide range of size (<1 to 2300 km²), maximum depth (3-117 m), mixing regimes, phytoplankton biomass and primary productivity (Table S1, Fig. S1). Phytoplankton biomass (Chlorophyll-a from 3.6 µg L$^{-1}$ to 190.2 µg L$^{-1}$) was dominated by Cyanobacteria (>95%) in the most productive lakes, while Diatoms (<20%) and Chrysophytes (<40%) also contributed in the less productive ones (Fig. S2). Maximum potential photosynthetic activity (P$_{max}$) varied from 1.5 µmol C L$^{-1}$ h$^{-1}$ in L. Edward to 199.0 µmol C L$^{-1}$ h$^{-1}$ in L. George and was linearly related to chlorophyll a concentration. Light-dependent N$_2$ fixation was detected in every lake with the exception of L. Kyamwinga. No significant N$_2$ fixation rates were measured in the dark. Maximum potential N$_2$ fixation rates (N$_2$fix$_{max}$) ranged between 1 nmol L$^{-1}$ h$^{-1}$ and 128 nmol L$^{-1}$ h$^{-1}$ and were positively related to P$_{max}$ (Fig. S1).

We detected and quantified the abundance of the archaeal alpha subunit of methyl-coenzyme M reductase gene (*mcrA*), a proxy for methanogens, in the surface waters of each lake. *mcrA* gene copy abundance (*mcrA* copy ng DNA$^{-1}$) ranged between 319 ± 41 (L. Edward) and 7537 ± 476 (L. Katinda) in the fraction of seston < 5 µm, and between 541 ± 19 (L. Edward) and 7968 ± 167 (L. Katinda) in the fraction of seston > 5 µm (Fig. S3). Illumina 16S rRNA gene amplicon sequencing indicated that methanogens accounted for a small fraction of the prokaryotic community in the surface waters of L. Edward (0.01 %), L. Kyamwinga (0.03 %) and L. Nyamunsingere (0.08 %). They represented a substantially higher fraction of the community in L. Katinda (0.38%) and L. George (0.57 %) (Fig. S4). In all lakes, hydrogenotrophic (*Methanomicrobiales* and *Methanobacteriales*) were always more abundant than acetoclastic (*Methanosarcinales*) microorganisms, representing at least 65% of the methanogens (up to 95% in L. Katinda, Fig. S4).

Surface waters were super-saturated in CH$_4$ in all lakes, with surface concentrations (at 1 m) ranging between 78 and 652 nmol L$^{-1}$ (atmospheric equilibrium ~ 2 nmol L$^{-1}$). Vertical patterns of CH$_4$ and stable carbon isotope composition of CH$_4$ ($\delta^{13}$C-CH$_4$) were variable among the different lakes. In L. Kyamwinga and Katinda, higher CH$_4$ concentrations and lower $\delta^{13}$C-CH$_4$ values were observed in the well-oxygenated epilimnion compared to the metalimnion showing a source of relatively $^{13}$C-depleted CH$_4$ to the epilimnetic CH$_4$ pool (Fig. 1). The CH$_4$ concentrations and $\delta^{13}$C-CH$_4$ were homogeneous in the water column of L. Edward that is much larger than the other studied lakes (2300 km², Table S1) and characterized by a higher wind exposure and a substantially weaker thermal stratification (Fig. 1). However, a clear horizontal gradient in CH$_4$ concentration and $\delta^{13}$C-CH$_4$ occurred between the littoral and pelagic zones (Fig. S5). Vertical gradients were also observed at much smaller scale in the near sub-surface (top 0.3 m) in the shallow and entirely well oxygenated L. George and L. Nyamusingere (Fig. 2). In both lakes CH$_4$ concentrations were relatively modest in the hypolimnion (< 50 nmol L$^{-1}$) but increased abruptly in the thermal gradient (0.3 m interval) to reach a surface maximum > 240 nmol L$^{-1}$ (Fig. 2). $\delta^{13}$C-CH$_4$ mirrored this pattern with





significantly lower values in surface than at the bottom of the water column indicating that a source of relatively $^{13}$C-depleted CH$_4$ contributed to the higher epilimnitic CH$_4$.

### 3.2. Occurrence of microbial CH$_4$ production in surface waters

Despite the prevalence of oxic conditions, $^{13}$C-labelling experiments revealed that CH$_4$ was produced in surface waters of each lake with the exception of L. Kyamwinga (Fig. 3). The kinetic of incorporation of NaH$^{13}$CO$_3$ into the CH$_4$ pool revealed that a substantially higher amount of CH$_4$ was produced from dissolved inorganic carbon (DIC) in illuminated waters, and this mechanism of CH$_4$ formation appears to be related to photosynthesis, as none or only modest quantities of CH$_4$ were produced from $^{13}$C-labelled DIC under darkness or when photosynthesis was inhibited by DCMU (Figs. 3a and S6). Furthermore, CH$_4$ production from DIC appeared strongly correlated (r² = 0.91) to the photosynthetic activity (Fig. 4a) and N$_2$ fixation rates (Fig 4b), supporting the view that CH$_4$ formation in oxic waters was directly linked to phytoplankton metabolism (Bizic et al. 2020).

Aside from DIC, an appreciable amount of CH$_4$ was generated in all lakes from the sulfur bonded methyl group of methionine when bottles were incubated under light, irrespective of the addition of DCMU (Fig. 3b and S6), that were approximately 4 times higher than in the dark. In addition, a positive relationship between CH$_4$ production from methionine in the light and the photosynthetic activity was found (Fig. 4c).

$^{13}$C-labelled acetate, the substrate of acetoclastic methanogenesis, supported the production of CH$_4$ in all lakes with the exception of L. Kyamwinga, but at much lower rates compared to light-dependent CH$_4$ production from DIC (50 times lower, n=7) or methionine (10 times lower, n=4) (Fig. 3c and S6). δ$^{13}$C analysis of the DIC in the bottles spiked with $^{13}$C-labelled acetate showed that the acetate was mineralized at rates of 5-6 orders of magnitude higher than acetoclastic methanogenesis so that added acetate appeared to be used almost exclusively by heterotrophic micro-organisms other than methanogens. Pattern of acetate-derived production of CH$_4$ were similar in light and dark treatments (Figs. 3c and S6) and this mode of CH$_4$ production appeared unrelated to phytoplankton activity (Fig. 4d).

### 3.3. Mechanisms of epilimnitic CH$_4$ production

Only a minimal fraction of the CH$_4$ produced under aerobic conditions originated from acetate in contrast with several earlier studies (Bogard et al. 2014, Donis et al. 2017) which proposed, based on the apparent fractionation factor of δ$^{13}$C-CH$_4$, that acetoclastic methanogenesis linked to phytoplankton production of organic matter would be the dominant biochemical pathway of pelagic CH$_4$ production in oxic freshwaters. Instead, our results suggest that epilimnetic CH$_4$ production in well-oxygenated conditions was related to DIC fixation by photosynthesis (Fig. 3), and correlated to primary production (Fig. 4a) and N$_2$ fixation (Fig 4b). When normalized to POC concentrations, the average DIC-derived CH$_4$ production rates (0.08 ± 0.05 nmol mmol$_{POC}$$^{-1}$ h$^{-1}$ n = 7) was remarkably similar to the CH$_4$ production rates recently reported in Cyanobacteria cultures (0.04 ± 0.02 nmol mmol$_{POC}$$^{-1}$ h$^{-1}$) grown at 30°C, among which the freshwater *Microcystis aeruginosa* (Bizic et al. 2020), the dominant Cyanobacterium species in the tropical lakes investigated in our study (see Fig S2). These CH$_4$ production rates are 2 orders of magnitude higher than rates reported in an axenic culture of the eukaryote *Emiliania huxleyi* (0.19 ± 0.07 pmol mmol$_{POC}$$^{-1}$ d$^{-1}$) (Lenhart et al. 2016), but they are 4 orders of magnitude lower than typical anoxic CH$_4$ production rates by methanogenic Archaea (Mountford & Asher 1979). Although it seems improbable that $^{13}$C-DIC acted as a direct precursor molecule for the CH$_4$ released by phytoplankton (Lenhart et al. 2016, Klintzsch et al. 2019) $^{13}$C-DIC could have been taken up by phytoplankton cells and then used as a C source for the synthesis of many different organic molecules that may serve as the actual CH$_4$ precursors. Indeed, healthy phytoplankton cells actively release a variety of low molecular weight molecules which are generally highly labile and rapidly consumed (Baines & Pace 1991, Morana et al. 2014). Phytoplankton metabolism could have fuelled CH$_4$ production pathways, at least partially, excreting substrates involved in CH$_4$ production via biochemical



processes such as demethylation of a variety of organic molecules like methionine, one of the S-bonded methylated amino acids (Lenhart et al. 2016), trimethylamine (Bizic et al 2018), or methylphosphonate (Yao et al. 2016).

While the source of methylphosphonate in freshwaters is obscure and its actual natural abundance remains to be determined, dissolved free amino acids would represent up to 4% of the DOC produced by phytoplankton and are rapidly consumed by heterotrophic bacteria (Sarmento et al. 2013). Our incubations indeed demonstrated that the methyl group of methionine was a potential precursor of $CH_4$ in all lakes investigated, in line with recent findings showing that *Emiliania huxleyi* could act as a direct source of $CH_4$ in oxic conditions using methionine as precursor, without involvement of any other

micro-organisms (Lenhart et al. 2016). We found that $CH_4$ production from methionine was clearly stimulated under light, even when photosynthetic activity was inhibited by DCMU, while little $CH_4$ from methionine was produced in darkness (Fig. 3b). DCMU notably prevents reduction of plastoquinone at photosystem II and generates singlet oxygen (Petrillo et al. 2014). The mechanism of $CH_4$ production from methionine is still unclear, but its residue in proteins is particularly sensitive to oxidation to methionine sulfoxide by radical oxygen species (ROS) (Levine et al. 1996) so that methionine would act as an

effective ROS scavenger and play important protective roles under photooxidative stress conditions, as shown in vascular plants (Bruhn et al. 2012). The side chain of methionine sulfoxide is identical to dimethyl sulfoxide which is known to react with hydroxyl radicals (OH) to form $CH_4$ (Repine et al. 1979). Besides its photoprotective role for phytoplankton, methionine could also be catabolized by a wide variety of microorganisms to methanethiol, which could in turn be transformed to $CH_4$ as shown in Arctic Ocean surface waters (Damm et al. 2010). Nevertheless, occurrence of this latter mechanism in the tropical

lakes investigated seems unlikely as this mode of $CH_4$ production would be expected to be insensitive to light irradiance and no $CH_4$ was produced from methionine in the dark during the incubations.

### 3.4. Relevance of epilimnitic $CH_4$ production compared to $CH_4$ loss terms at ecosystem scale

Net $CH_4$ oxidation was detected in all 5 investigated lakes ranging from 11 to 5212 nmol $L^{-1}$ $d^{-1}$ (Fig. 5), and was by far the largest loss term of dissolved $CH_4$ at ecosystem scale (8 to 46 times higher than the diffusive emission to the

atmosphere). Surface water $CH_4$ turnover times were particularly short in the shallow and eutrophic L. George (2h) and L. Nyamusingere (3h) and slightly longer in the deeper and less productive L. Katinda (11h), L. Kyamwinga (77h) and L. Edward (100h). In all studied lakes, pelagic $CH_4$ production rates measured during the stable isotope tracer experiments represented between 0.1% and 8.5% of net $CH_4$ oxidation rates, regardless of the $CH_4$ precursors tested.

All of the major sources and sinks of $CH_4$ at ecosystem scale were experimentally determined offshore in three lakes

(L. Edward at 20 m depth, George and Nyamusingere) (Fig. 6). In these three lakes, surface $CH_4$ production rates from the diverse precursors molecules investigated were modest relative to the diffusive $CH_4$ efflux to the atmosphere (0.4 – 20.0 %) and microbial $CH_4$ oxidation (0.1 – 13.2 %). In opposition, the combined $CH_4$ bubble dissolution flux and diffusive benthic $CH_4$ flux were several orders of magnitude higher than $CH_4$ production in surface waters, and met the $CH_4$ loss terms (emission and oxidation) (Fig. 6). These results gathered in tropical lakes of various size (from 0.44 to 2300 km²) and depth are in sharp

contrast with the estimation of an empirical model (Gunthel et al. 2019) which proposed that mechanisms of oxic $CH_4$ production represents the majority of $CH_4$ emissions in lakes larger than 1 km². This discrepancy highlights the need to consider the unique limnological characteristics of a vast region of the world that harbours 16% of the total surface of lakes (Lehner & Doll 2004). One of the most distinctive features of tropical aquatic environment is the persistent elevated water temperature in the hypolimnion and at the water-sediment interface which favours methanogenic activity in sediment and decreases $CH_4$

solubility, enhancing bubbles formation.

### 3.5. Origin of $^{13}C$ depleted $CH_4$ in surface waters

Epilimnetic $CH_4$ production was a marginal flux at ecosystem scale and could not explain alone the accumulation of $^{13}C$-depleted $CH_4$ in the epilimnion of most of the lakes of our dataset (Figs. 1, 2), for which we propose two other alternative





mechanisms: dissolution of arising $CH_4$ bubbles in the epilimnion combined with inhibition by light of $CH_4$ oxidation. The partial dissolution of the $CH_4$ bubbles as they rise in the epilimnion should allow a rapid transport of $^{13}C$-depleted $CH_4$ from the sediment, bypassing the hotspot of $CH_4$ oxidation at the sediment-water interface and representing an alternative source of $^{13}C$-depleted $CH_4$ in water column. The shallower L. George and L. Nyamusingere were notably characterized by sharp thermal density gradients (Fig. 2) and extreme phytoplankton biomass largely dominated by *Microcystis aeruginosa* (Chlorophyll a up to 190 µg $L^{-1}$). *Microcystis aeruginosa* cells form large aggregates (>1 mm) embedded in a matrix of extracellular polymeric

substance that might act as a barrier to trap small $CH_4$ bubbles arising from the sediment (Fig S7). Dissolution of $CH_4$ bubbles could be enhanced at the very near surface due to the entrapments of bubbles at the air-water interface by abundant surface organic films that delay the bubble "burst". The presence of a sharp sub-surface temperature gradient would further enhance $CH_4$ accumulation during day-time near the air-water interface (by blocking vertical redistribution of $CH_4$ by mixing). We hypothesize that this process could be widespread in shallow tropical lakes which are characterized by high productivity and

are susceptible to be simultaneous large benthic $CH_4$ sources.

The influence of light on methanotrophy was investigated in the deep L. Edward, and shallow L. George and L. Nyamusingere, revealing that $CH_4$ oxidation rates decreased dramatically with increasing light intensity (Fig. 5). For instance, when exposed to full sunlight intensity, methanotrophs consumed only 42% (L. Edward), 54% (L. Nyamusingere) or 74% (L. George) of the $CH_4$ they were able to oxidize in the dark. The magnitude of this light-induced inhibition decreased substantially

with decreasing sunlight intensities, as shown elsewhere (Murase & Sugimoto 2005). The physiological mechanism of photoinhibition of $CH_4$ oxidation could be related to the fact that the copper-containing methane monooxygenase enzyme and structurally close to the ammonia monooxygenase enzyme, and might be inactivated by ROS produced during photooxidative stress, as shown for ammonium oxidizers (French et al. 2012, Tolar et al. 2016). Altogether, our results emphasize the role of sunlight irradiance as an important, but frequently overlooked, environmental factor driving the $CH_4$ dynamics in lake surface

waters, and possibly contributing to the occurrence of $^{13}C$ depleted $CH_4$ in surface waters.

### 4. Supplement

Supplementary figures are available on-line on the *Biogeosciences* website.

### 5. Data availability

All data included in this study are available upon request by contacting the corresponding author.

### 6. Author contributions

This study was designed by C. Morana, A.V. Borges & S. Bouillon. All authors participated to samples collection, data acquisition and analysis, and to the drafting of the manuscript. All authors approved the final version of the manuscript.

### 7. Competing interests

The authors declare that they have no conflict of interest.

### 8. Acknowledgments

We are grateful to Marc-Vincent Commarieu and Dries Grauwels for their help in the lab and during the field sampling. We also thank the Uganda Wildlife Agency for research permission in the Queen Elizabeth National Park (Uganda), the staff of the Tembo Canteen for the use of their incubation room and the crew of the Katwe Marine Research Vessel for their help during the L. Edward sampling. This work was funded by the Belgian Federal Science Policy Office (BELSPO, HIPE project,
BR/154/A1/HIPE) and by the Fonds Wetenschappelijk Onderzoek (FWO-Vlaanderen, Belgium) with travel grants awarded to CM and SB. AVB is a research director at the Fond National de la Recherche Scientifique (FNRS, Belgium).

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

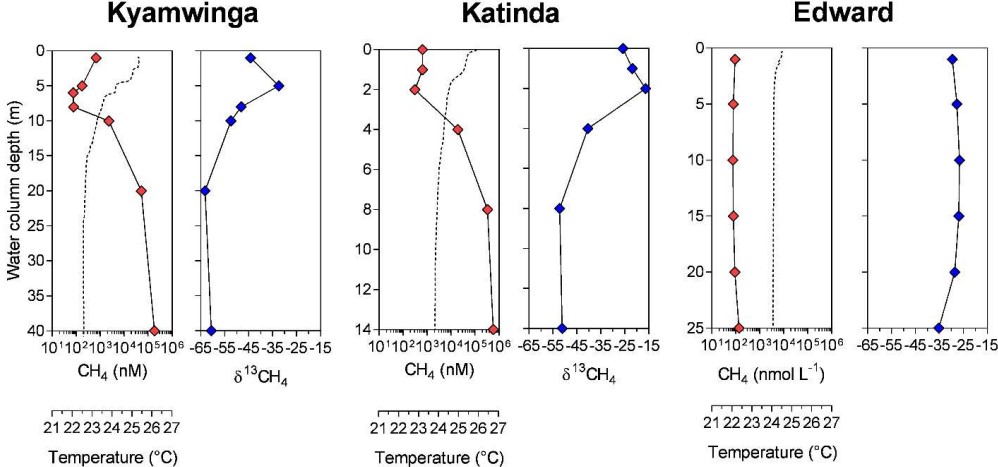

**Figure 1. Depth profile.** Depth profile of the temperature (°C ; dashed line), $CH_4$ concentration (nmol $L^{-1}$ ; red symbols) and stable isotope carbon composition of $CH_4$ ($\delta^{13}$C-$CH_4$, ‰ ; blue symbols) in Lake Kyamwinga (left), Lake Katinda (middle),
and Lake Edward (right).






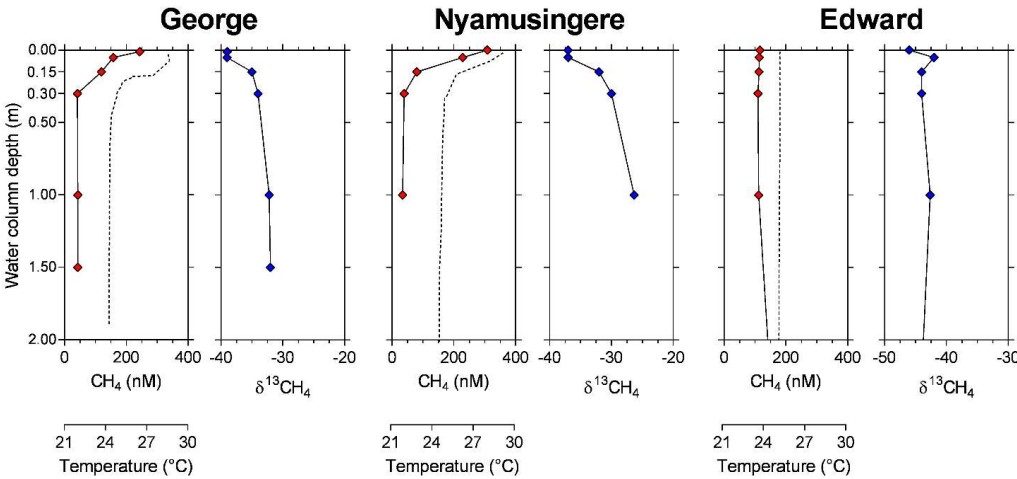

**Figure 2. Depth profile, focus on the surface.** Depth profile of the temperature (°C ; dashed line), $CH_4$ concentration (nmol $L^{-1}$ ; red symbols) and stable isotope carbon composition of $CH_4$ ($\delta^{13}C$-$CH_4$, ‰ ; blue symbols) in Lake George (left), Lake Nyamusingere (middle), and the surface waters (0-2 m) of Lake Edward (right).





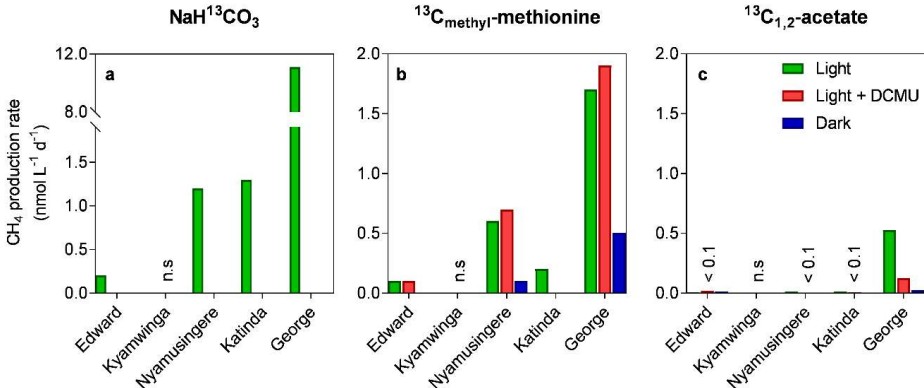


**Figure 3. Tracer experiments show CH$_4$ production in well-oxygenated surface waters.** CH$_4$ production rates (nmol L$^{-1}$ d$^{-1}$) from dissolved inorganic carbon (a), the methyl group of methionine (b) and acetate (c) measured in the surface waters (0.3 m) of a variety of African tropical lakes. Green, grey and dark bars respectively represent rates measured under light, light in presence of a photosynthesis inhibitor (DCMU), or darkness. Values showed for L. Edward, L. George and L. Katinda are
the average of 2017 and 2018 sampling campaign measurement. n.s = not significant, < 0.1 = below 0.1 nmol L$^{-1}$ d$^{-1}$.







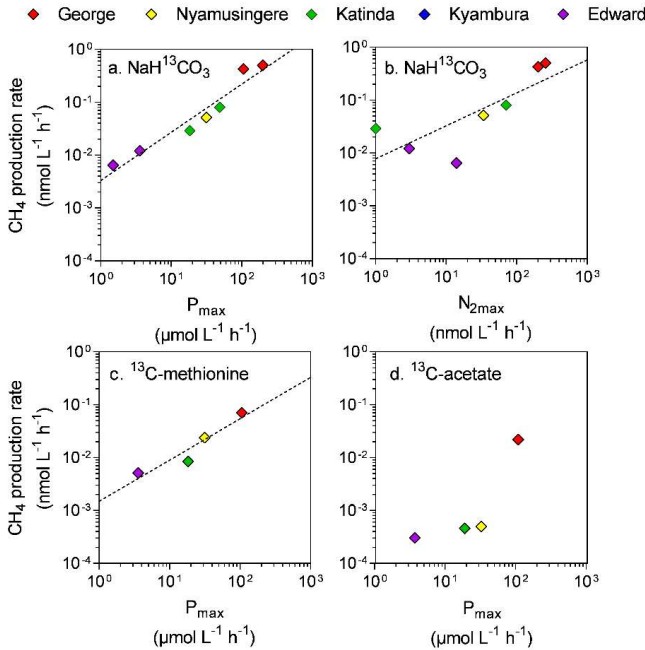

**Figure 4. Direct link between CH₄ production and phytoplankton metabolism.** Relationship between the maximum photosynthetic activity ($P_{max}$, µmol C L$^{-1}$ h$^{-1}$) or maximum nitrogen fixation rates ($N_{2max}$, nmol L$^{-1}$ h$^{-1}$) and surface CH₄ production rates (nmol C L$^{-1}$ h$^{-1}$) from dissolved inorganic carbon (a, b), methyl group of methionine (c), and acetate (d).







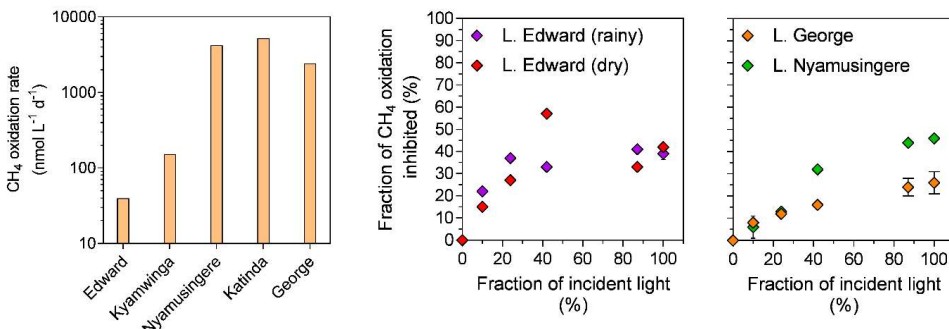

**Figure 5. Light inhibition patterns of CH₄ oxidation in surface waters.** Left panel: CH$_4$ oxidation rates (nmol L$^{-1}$ d$^{-1}$) measured in the surface waters (0.3 m) in the dark of a variety of African tropical lakes. Right panel: relationship between illumination (fraction of incident sunlight irradiance, %) and CH$_4$ oxidation inhibition (fraction of CH$_4$ oxidation in the dark inhibited at a given irradiance, %) in Lake Edward, Lake George and Lake Nymusingere. Symbols represent the mean, and error bars represent the maximum and minimum of duplicate experiments




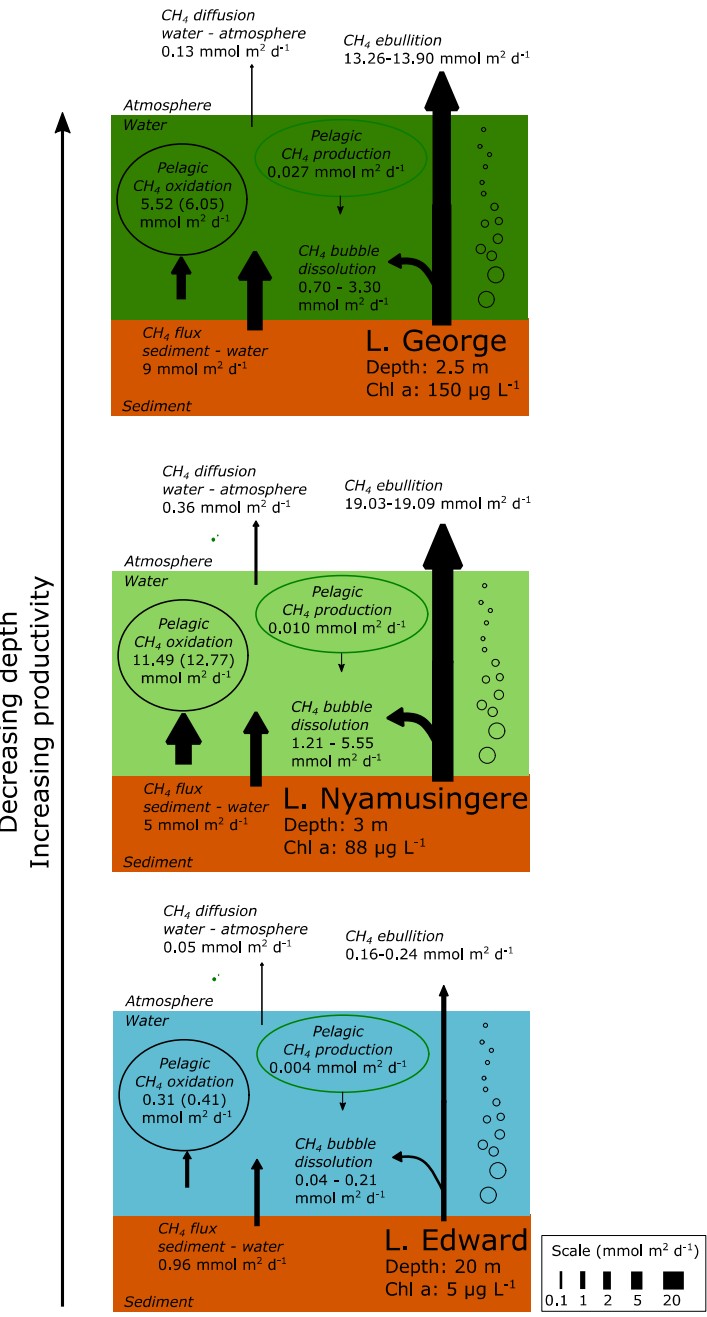

**Figure 6. Epilimnitic CH₄ production is a marginal source of CH₄ compared to sedimentary sources and CH₄ sinks in several contrasting African lakes.** Summary of the different $CH_4$ flux experimentally measured in L. Edward, L. George and L. Nyamusingere. Values of $CH_4$ oxidation in brackets are values not considering $CH_4$ photoinhibition. Pelagic $CH_4$ production are values determined from $NaH^{13}CO_3$ (~5% final enrichment) and $^{13}C$-acetate (99% final enrichment), as described in the methods section. $^{13}C$-labelling experiment carried out under constant light irradiance. $CH_4$ flux at the water-air and sediment-water interface were determined experimentally as described in the Methods. $CH_4$ bubble dissolution and $CH_4$ ebullition flux





were determined using the SiBu-GUI software (Greinert & McGinnis 2009); minimum and maximum represents the values obtained from two extreme bubble-size scenarios considering a release of many small (3 mm diameter) bubbles or fewer large (10 mm) bubbles.




