# Peer review of "Methane paradox in tropical lakes? Sedimentary fluxes rather than pelagic production in oxic conditions sustain methanotrophy and emissions to the atmosphere"

_Biogeosciences, 2020_

## Referee Comment (RC1) · Anonymous Referee #1 · 27 May 2020

In their manuscript, submitted to Biogeosciences, the authors present data collected on five African lakes with the goal to quantify methane production in oxic waters and the other relevant sources and sinks. Their conclusion is that methane produced in the anoxic sediment and released via diffusive flux or ebullitive fluxes is the primary source of methane to the surface water. They state that empirical models for pelagic methane production (PMP) do not account for tropical systems as they were developed from temperate and often meso- or oligotrophic lakes. Their work ultimately aims to assess the importance of surface water methane production in oxic conditions by determining

the relevant sources – namely PMP via incubations/tracer experiments, diffusive and bubble methane release from the sediment, air-water transport and oxidation.

While I agree with the authors' ideas that trophic lakes need to be carefully considered and are likely not comparable to higher-latitude lakes, there are some areas I am unclear in within their manuscript. I find the paper hard to follow in several places, and the structure could be improved. I am not sure if combining the results and discussion is the best approach, as results are often buried in the text, and the logic becomes confusing to follow. Furthermore, the lakes are not always listed in the same order in the tables and figures, making it more cumbersome to compare separate results on the same lake. Finally the "mass balance" presented on Fig. 6 is strange, and hard to follow. In fact the balances do not close with the rates presented.

Finally, while the authors clearly did a vast amount of excellent work on these lakes, and presented very intriguing data, there needs to be a better assessment of the large uncertainty in the analysis and methodology, clearer presentation of the data, and locations where the samples were obtained (i.e. maps with sample locations are a must). Again, these are very interesting data, and the methane varies between the lakes in some still unexplained ways. Perhaps a closer comparison of the lakes and their properties in relation to methane concentrations, d13C signatures, etc. I list the individual (both minor and major) comments/questions below more-or-less in order they appear in the text.

- Environmental Setting: In general, please define where the samples were obtained. I think a map of each lake with the location would be extremely helpful. For example, there is no indication where profiles were obtained other than depth. Furthermore we are missing the locations of the sediment obtained for the sediment-water flux determinations. Finally, I think the oxygen profiles should be included on figures 1 and 2.

- Diffusive flux at air-water interface: This is always an area of controversy and uncertainty. I suggest utilizing several parameterizations, perhaps some more recent, to at least give a range. I could suggest e.g. (MacIntyre et al. 2010; Vachon et al. 2010), or additional/others. Furthermore, how close was the weather station to the lake sampling points.

- Ebullition flux: How were the locations selected for the bubble flux measurements with the funnels? What was the assumed %methane of the initial bubble gas as a significant portion of the bubble gas from shallow bubble release is N2 (Langenegger et al. 2019). Finally how were bubble sizes selected? I believe some literature values are available.

- CH4 flux across the sediment-water area: I like the method the authors' used here. The main issue I have here is that because the incubations were performed after the removal of oxygen, the flux rate they get may be on the very upper end. Perhaps this should be viewed as a potential maximum methane flux. By removing the oxygen, the methane oxidizing layer at the sed-water interface was removed resulting in artificially large fluxes. This has been shown in several instances – see e.g. (Damgaard et al. 1998; Liikanen and Martikainen 2003)(Liikanen Fig 2).

- CH4 oxidation rates: Did these incubations remain oxic throughout? I find the rates reported rather on the upper end of reported values. It would be useful to compare your measured oxidation rates with literature values – especially for tropical lakes.

- Sunlight inhibitory effect on methane oxidations: Here, I have the same question regarding oxygen concentrations in the bottles. Furthermore, the authors state that they "investigate the hypothetical inhibitory effect of dissolved O2 production [on methane oxidation] by phytoplankton" however this was never discussed again. Since O2 was not reported, could the effects they see with reduction of oxidation with increasing light exposure rather be related to oxygen concentrations? Finally, as I understand, serum bottles block considerable light from penetrating, how is this considered?

- Determination of pelagic methane production: Here I admit I am not an expert. The authors use DCMU to inhibit photosynthesis. However, is it not important that the

methane oxidation is inhibited with methyl fluoride? In other words, with the high reported oxidation rates, how does the oxidation that occurs within the incubations accounted for? Finally how did you ensure that the samples remained oxic through the experiment?

- Mass balance (figure 6): I think the mass balances are slightly misleading. Firstly, L. George is missing a source of 6 mmol/m2/d to close the balance, while the oxidation rate is very to high in L. Nya to close the balance. At any rate, such a mass balance would need to be performed over the lake scale. However, given the very limited data such a mass balance would also have a large amount of uncertainty. I suggest putting this information into a table and be very detailed that these are point measurements over a very large lake and thus may not be representative of the overall conditions. Please list uncertainties in these estimates.

- As an example for the L. George on figure 6. In mmol/m2/d the sources are Sed + bubble + PMP which is $9 + \sim2 + 0.027 = \sim11$. The losses are oxidation + atm = $5.5 + 0.13 = \sim5.5$. If this is meant to be a mass balance, and assuming steady state, there is a missing source term of 5.5 mmol/m2/d.

Damgaard, L. R., N. P. Revsbech, and W. Reichardt. 1998. Use of an Oxygen-Insensitive Microscale Biosensor for Methane To Measure Methane Concentration Profiles in a Rice Paddy. Appl. Environ. Microbiol. 64: 864-870. Langenegger, T., D. Vachon, D. Donis, and D. F. McGinnis. 2019. What the bubble knows: Lake methane dynamics revealed by sediment gas bubble composition. Limnol Oceanogr 0. Liikanen, A., and P. J. Martikainen. 2003. Effect of ammonium and oxygen on methane and nitrous oxide fluxes across sediment-water interface in a eutrophic lake. Chemosphere 52: 1287-1293. MacIntyre, S., A. Jonsson, M. Jansson, J. Aberg, D. E. Turney, and S. D. Miller. 2010. Buoyancy flux, turbulence, and the gas transfer coefficient in a stratified lake. Geophys. Res. Lett. 37. Vachon, D., Y. T. Prairie, and J. J. Cole. 2010. The relationship between near-surface turbulence and gas transfer velocity in freshwater systems and its implications for floating chamber measurements of gas exchange.

Limnol. Oceanogr. 55: 1723-1732.

---

## Referee Comment (RC2) · Anonymous Referee #2 · 2 Jul 2020

In their paper, the authors undertake and extremely comprehensive set of measurements to assess the methane paradox in freshwater lakes. The authors are to be commended for such a comprehensive set of experiments, in what must have been difficult environments to work in.

Overall I found the manuscript well written, and the data supported the conclusions raised. I would suggest that some parts be toned down however, due to the (understandable) lack of replication spatially and temporally. For example, the mass balance calculations are derived from short term experiments/measurements with restricted

spatial replication. While this in itself is not a terminal flaw, I think a more nuanced assessment of the results is required. I certainly appreciate the trade-off with doing a large number of experiments and measurements over a range of systems, versus long term intensive experiments on a single system.

I would also suggest separating results and discussion to simplify the narrative,this would improve the readability of the paper, and also prevent some of the interesting findings being lost in a sea of descriptive text.

Specific comments: Line 18 Dissolution flux was modeled rather than measured right?

Line 46 "Among others", reword to clarify

Section 2.5 I appreciate that measuring benthic fluxes of CH4 are difficult, but I wonder how represenattive these core experiments are to insitu rates. The cores had water drained, what affect might this have on the microbial community (i.e. introducing O2 into sediments). Further, the shallow sediment depth may also introduce artifacts. Is there any information on sediment characteristics that may help the reader to intepret the potential issues associated with this (e.g. porosity etc.). Further, are bottom waters anoxic in the lakes (as the water used for incubations was anoxic).

Line 186 Would the method used for d13C-DIC measurement also pick up any labeled 13C-CH4? I would expect that the EA-IRMS method would oxidize CH$ to CO2 and that this may introduce an artifact, but maybe I missed something with the method description.

Line 200 Was ambient concentrations of ambient acetate and methionine measured or just estimated?

Line 313 "which was"

Line 357 "of the vast tropical region"

Once again - congratulations to the authors on a very nice study.

---

## Author Comment (AC1) · 5 Aug 2020

**Reviewer 1 comments (bg-2020-142)**

We thank the reviewer for his/her careful reading of the manuscript and for providing helpful and thoughtful comments. We provide below a point-by-point reply to the comments. Reviewer comments are italicized while our responses are not italicized.

*Reviewer: While I agree with the authors' ideas that tropical lakes need to be carefully considered and are likely not comparable to higher-latitude lakes, there are some areas I am unclear in within their manuscript. I find the paper hard to follow in several places, and the structure could be improved. I am not sure if combining the results and discussion is the best approach, as results are often buried in the text, and the logic becomes confusing to follow. Furthermore, the lakes are not always listed in the same order in the tables and figures, making it more cumbersome to compare separate results on the same lake. Finally, the "mass balance" presented on Fig. 6 is strange, and hard to follow. In fact, the balances do not close with the rates presented.*

Reply: We followed the reviewers' suggestion and changed considerably the structure of the manuscript. Results and discussion are now split in two different sections. We would also like to point that the figure 6 does not depict the results of a mass balance but is instead a simple graphical illustration of the different fluxes measured independently. Actually, the words "mass balance" are not mentioned in the manuscript, we instead described the figure 6 as a "summary of the different $CH_4$ flux experimentally measured in L. Edward, L. George and L. Nyamusingere" (Line 627). Due to the empirical nature of the values reported in Figure 6 and the uncertainties around every measurement, we were not expecting to be able to bring a closed mass balance. Instead, the main purpose of the figure 6 was to illustrate the large discrepancy between the pelagic $CH_4$ production and the $CH_4$ oxidation and $CH_4$ emission fluxes.

*Reviewer: Finally, while the authors clearly did a vast amount of excellent work on these lakes, and presented very intriguing data, there needs to be a better assessment of the large uncertainty in the analysis and methodology, clearer presentation of the data, and locations where the samples were obtained (i.e. maps with sample locations are a must). Again, these are very interesting data, and the methane varies between the lakes in some still unexplained ways. Perhaps a closer comparison of the lakes and their properties in relation to methane concentrations, $d^{13}C$ signatures, etc. I list the individual (both minor and major) comments/questions below more-or-less in order they appear in the text.*

Reply: We thank the reviewer for his/her positive evaluation of our work. We believe that the improved structure of the revised version of the manuscript (split results and discussion sections) allow a clearer presentation of the data and a closer comparison of the $CH_4$ dynamic between the lakes, as requested by the reviewer. See reply to the comment below regarding the map.

*Reviewer : Environmental Setting: In general, please define where the samples were obtained. I think a map of each lake with the location would be extremely helpful. For example, there is no indication where profiles were obtained other than depth. Furthermore, we are missing the locations of the sediment obtained for the sediment-water flux determinations. Finally, I think the oxygen profiles should be included on figures 1 and 2.*

Reply: The coordinate of the sampling sites can be found in the material and method. We unfortunately don't have a detailed map of the lakes we sampled (at the exception of L. Edward) but we think that an interested reader could easily visualize the location of each sampling station using the coordinates we provide and a widely available software such as Google Earth. Nevertheless, we modified the site

description section of the material and method to provide the distance to shore and the water column depth of every sampling site. Sediment water fluxes were measured at the same location where processes and air-water fluxes were measured (coordinates are given in the material and methods). The oxygen profiles were added to the figures 1 and 2 following the reviewer suggestion.

***Reviewer : Diffusive flux at air-water interface: This is always an area of controversy and uncertainty. I suggest utilizing several parameterizations, perhaps some more recent, to at least give a range. I could suggest e.g. (MacIntyre et al. 2010; Vachon et al. 2010), or additional/others. Furthermore, how close was the weather station to the lake sampling points***

Reply : Coordinates of the sampling sites and of the position of the weather station are given in the material and methods. We acknowledge that parametrization of the gas transfer velocity is controversial, but the model we used here (Cole and Caraco 1998) is by far the most widely applied in the literature, which we believe will facilitate comparison with other studies. It is also one of the simplest because it allows to parametrize the gas transfer velocity as a function of wind speed alone. Its simplicity is an advantage in studies carried out in remote location such Western Uganda where access to sophisticated weather station is not possible. Furthermore, a recently published paper (Klaus & Vachon 2020, Aquatic Sciences) compared the performance of several wind based empirical model and concluded that the model of Cole and Caraco (1998) we used do not perform differently (better or worse) than other (including the model of MacIntyre et al. 2010 and Vachon et al. 2010 proposed by the reviewer).

***Reviewer : Ebullition flux: How were the locations selected for the bubble flux measurements with the funnels? What was the assumed %methane of the initial bubble gas as a significant portion of the bubble gas from shallow bubble release is $N_2$ (Langenegger et al. 2019). Finally how were bubble sizes selected? I believe some literature values are available.***

Reply : The ebullition was measured at the same sampling sites where we performed the other measurements. Water depth of the sampling site was 20 m, 2.5 m and 3 m for L. Edward, George and Nyamusingere, respectively, as now explained in the material and methods. We acknowledge that ebullition is strongly variable in function of the water column depth and hence we expect this flux to show large spatial variation in a deep lake such as L. Edward, with a maximal depth of 117 m and a mean depth of only 34 m. This important element has been added in the discussion section. However, L. George and L. Nyamusingere are shallow lakes with a rather homogeneous bathymetry (maximal depth of 7m for a mean depth of 2.5 m /3 m) so that the ebullition measurement we performed at 2.5 and 3 m could be extrapolated to the entire lake.

The initial fraction of $CH_4$ contained in the arising bubble was calculated back for every bubble size scenario using the Sibu-GUI software (Greinert & McGinnis 2009) from the measured fraction of $CH_4$ in the bubbles trapped in the funnel, the temperature, and the bubble release depth (equivalent to the sampling site depth). The rest of the gas was assumed to be $N_2$. Bubble dissolution depends largely on bubble size, we then chose to consider ebullition following 3 bubble size scenarios, as explained in the material and methods (3 mm, 6 mm, 10 mm). These values were selected because a previous work (Delwiche & Hemond 2017) showed the vast majority of the bubble released from sediment were from this size interval. This reference (Delwiche & Hemond 2017) has been added in the manuscript.

***Reviewer : $CH_4$ flux across the sediment-water area: I like the method the authors' used here. The main issue I have here is that because the incubations were performed after the removal of oxygen, the flux rate they get may be on the very upper end. Perhaps this should be viewed as a potential maximum methane flux. By removing the oxygen, the methane oxidizing layer at the sed-water***

*interface was removed resulting in artificially large fluxes. This has been shown in several instances – see e.g. (Damgaard et al.1998; Liikanen and Martikainen 2003) (Liikanen Fig 2).*

Reply : This is correct and we thank the reviewer for raising this point. Aerobic $CH_4$ oxidation in the uppermost part of the sediment has indeed been probably inhibited following the removal of $O_2$. This will be clarify in the material and method of the revised manuscript. The term "$CH_4$ flux across the sediment water interface" will also be changed to "potential $CH_4$ flux across the sediment interface", following the reviewer suggestion.

*Reviewer : $CH_4$ oxidation rates: Did these incubations remain oxic throughout? I find the rates reported rather on the upper end of reported values. It would be useful to compare your measured oxidation rates with literature values – especially for tropical lakes.*

Reply : Yes, O2 consumption was measured in incubation bottles during a parallel experiment and the results showed the water remain oxic during the course of the experiment. This important observation is now mentioned in the text. As explained in the material and methods, the methane oxidation bottles were actually incubated during a relatively short time period to avoid anoxia (maximum 24h, but only 6h in the eutrophic L. George and Nyamusingere)

*Reviewer :* **Sunlight inhibitory effect on methane oxidations: Here, I have the same question regarding oxygen concentrations in the bottles. Furthermore, the authors state that they "investigate the hypothetical inhibitory effect of dissolved O2 production [on methane oxidation] by phytoplankton" however this was never discussed again. Since O2 was not reported, could the effects they see with reduction of oxidation with increasing light exposure rather be related to oxygen concentrations? Finally, as I understand, serum bottles block considerable light from penetrating, how is this considered?**

Reply : The bottles remained oxic during the full course of the incubation, see reply to the previous comment.

Indeed, a preliminary experiment was carried out in L. Edward and L. George only to investigate the hypothetical inhibitory effect of dissolved O2 production by phytoplankton. These results were missing in the previous version of the manuscript but are now briefly presented and discussed. They showed that $CH_4$ oxidation followed the same pattern of lower rates at high sunlight intensities regardless of DCMU addition.

Finally, all samples were incubated in the same type of borosilicate glass serum bottles so that we can assume that any sunlight attenuation caused by the bottles was identical for every sample and would not have affected the observed pattern of lower $CH_4$ rates at high sunlight intensities.

*Reviewer :* **Determination of pelagic methane production: Here I admit I am not an expert. The authors use DCMU to inhibit photosynthesis. However, is it not important that the methane oxidation is inhibited with methyl fluoride? In other words, with the high reported oxidation rates, how does the oxidation that occurs within the incubations accounted for? Finally how did you ensure that the samples remained oxic through the experiment?**

Reply : Photosynthesis (and then O2 production) occurred in the samples incubated under light, and O2 consumption in samples incubated under darkness was measured in parallel experiment and showed that oxic conditions remained during the full course of the incubation (see comments above). The $CH_4$ concentration in the incubation bottle was measured at every time step and the equation used to calculate the $CH_4$ production rate (equation 7) allowed to consider the evolution of the $CH_4$

concentration. In other words, any significant decrease of the $CH_4$ concentration due to $CH_4$ oxidation during the experiment was accounted in the calculation.

*Reviewer : Mass balance (figure 6): I think the mass balances are slightly misleading. Firstly, L.George is missing a source of 6 mmol/m2/d to close the balance, while the oxidation rate is too high in L. Nya to close the balance. At any rate, such a mass balance would need to be performed over the lake scale. However, given the very limited data such a mass balance would also have a large amount of uncertainty. I suggest putting this information into a table and be very detailed that these are point measurements over a very large lake and thus may not be representative of the overall conditions. Please list uncertainties in these estimates. As an example for the L. George on figure 6. In mmol/m2/d the sources are Sed +bubble + PMP which is 9 + ~2 + 0.027 = ~11. The losses are oxidation + atm = 5.5 +0.13 = ~5.5. If this is meant to be a mass balance, and assuming steady state, there is a missing source term of 5.5 mmol/m2/d.*

Reply : The figure 6 does not depict the results of a mass balance but is instead a simple graphical illustration of the different fluxes measured independently. Actually, the words "mass balance" are not mentioned in the manuscript, we instead described the figure 6 as a "summary of the different $CH_4$ flux experimentally measured in L. Edward, L. George and L. Nyamusingere" (Line 627). Due to the empirical nature of the values reported in Figure 6 and the uncertainties around every measurement, we were not expecting to be able to bring a closed mass balance. Instead, the main purpose of the figure 6 was to illustrate the large discrepancy between the pelagic $CH_4$ production and the $CH_4$ oxidation and $CH_4$ emission fluxes. We also modified the discussion to highlight the fact that ebullition in L. Edward was measured at a site of 20 m and may thus not be representative of the entire lake, as requested by the reviewer.

---

## Author Comment (AC2) · 5 Aug 2020

**Reviewer 2**

We thank the reviewer for his/her positive assessment of our work and sincerely appreciate his/ compliments. We provide below a point-by-point reply to the comments. Reviewer comments are italicized while our responses are not italicized.

*Reviewer : In their paper, the authors undertake and extremely comprehensive set of measurements to assess the methane paradox in freshwater lakes. The authors are to be commended for such a comprehensive set of experiments, in what must have been difficult environments to work in. Overall, I found the manuscript well written, and the data supported the conclusions raised. I would suggest that some parts be toned down however, due to the (understandable) lack of replication spatially and temporally. For example, the mass balance calculations are derived from short term experiments/measurements with restricted spatial replication. While this in itself is not a terminal flaw, I think a more nuanced assessment of the results is required. I certainly appreciate the trade-off with doing a large number of experiments and measurements over a range of systems, versus long term intensive experiments on a single system. I would also suggest separating results and discussion to simplify the narrative, this would improve the readability of the paper, and also prevent some of the interesting findings being lost in a sea of descriptive text.*

Reply : We followed the reviewers' suggestion and split the results and in two different sections. We hope it will improve the readability of the paper. We also took care to tone down the last part of the discussion.

*Reviewer : Specific comments: Line 18 Dissolution flux was modeled rather than measured right?*

Reply : Indeed, we agree with the reviewer. The sentence has been corrected.

*Reviewer : Line 46 "Among others", reword to clarify*

Reply : The sentence has been changed. It now reads "Primary production, methanogenic and methanotrophic activities, and cyanobacterial dominance are potentially much higher in tropical lakes due to favorable temperature (Lewis 1987, Kosten et al. 2012)".

*Reviewer : Section 2.5 I appreciate that measuring benthic fluxes of $CH_4$ are difficult, but I wonder how represenattive these core experiments are to insitu rates. The cores had water drained, what affect might this have on the microbial community (i.e. introducing O2 into sediments). Further, the shallow sediment depth may also introduce artifacts. Is there any information on sediment characteristics that may help the reader to intepret the potential issues associated with this (e.g. porosity etc.). Further, are bottom waters anoxic in the lakes (as the water used for incubations was anoxic).*

Reply : We would like to clarify that only the overlying water was drained. We also took care to avoid sediment disturbance using a tube connected to a 50 ml luer syringe to drain the water. Overlying water was then replaced by Helium-purged filtered (0.2 μm) water which was previously collected at the lake bottom. We are then confident with the fact that $O_2$ wasn't introduced in the sediments. The method description might have been confusing, it has been modified to improve its clarity.

However, "natural" bottom lake water was indeed oxic, while our incubation was carried out under anoxia. Aerobic $CH_4$ oxidation in the uppermost part of the sediment might have been inhibited following the removal of $O_2$. This will be clarified in the material and method of the revised manuscript. The term "$CH_4$ flux across the sediment water interface" will also be changed to "potential $CH_4$ flux across the sediment interface".

*Reviewer : Would the method used for d$^{13}$C-DIC measurement also pick up any labeled $^{13}$C-CH$_4$? I would expect that the EA-IRMS method would oxidize CH$ to CO2 and that this may introduce an artifact, but maybe I missed something with the method description.*

Reply : The setup of the EA-IRMS we used for the d$^{13}$C-DIC was modified as described in Gillikin & Bouillon (2007). Briefly, we installed an injection port in the Helium carrier gas line between the reduction column of the EA and the water trap. Since the sample gas is injected after the two EA furnaces (but before the water trap and the chromatography column), a contamination of the d$^{13}$C-DIC by some $^{13}$C-labelled CH$_4$ is impossible. A reference to the paper of Gillikin & Bouillon 2007 has been added in the revised version of the manuscript in order to clarify our method.

*Reviewer : Line 200 Was ambient concentrations of ambient acetate and methionine measured or just estimated?*

Reply : Ambient acetate was assumed to be 1 µmol L$^{-1}$ or lower based on literature values (Allen 1968 Ho et al. 2002, Tang et al. 2014). Final concentration of acetate in the bottles spiked with $^{13}$C labelled acetate was estimated at 100 µmol L$^{-1}$) Methionine was assumed to be lower than 0.1 µmol L$^{-1}$ (Sarmento et al. 2013). Final concentration of methionine in the bottles spiked with $^{13}$C labelled methionine was 10 µmol L$^{-1}$.